# Representation of fire, land-use change and vegetation dynamics in the Joint UK Land Environment Simulator Vn4.9 (JULES)

Chantelle Burton[1,2], Richard Betts[1,2], Manoel Cardoso[3], Ted R. Feldpausch[2], Anna Harper[2], Chris Jones[1], Douglas I. Kelley[4], Eddy Robertson[1], Andy Wiltshire[1]

5  [1]Met Office Hadley Centre, Exeter, UK. EX1 3PB, UK
[2]College of Life and Environmental Science, University of Exeter, Exeter. EX4 4SB. UK
[3]Brazilian Institute for Space Research (INPE), Earth System Science Center (CCST), São José dos Campos, Brazil
[4]Centre for Ecology and Hydrology, Wallingford. OX10 8BB. UK

10  *Correspondence to*: Chantelle Burton (chantelle.burton@metoffice.gov.uk)

## Abstract

Disturbance of vegetation is a critical component of land cover, but is generally poorly constrained in land-surface and carbon cycle models. In particular, land-use change and fire can be treated as large-scale disturbances without full representation of their underlying complexities and interactions. Here we describe developments to the land surface model JULES (Joint UK Land Environment Simulator) to represent land-use change and fire as distinct processes which interact with simulated vegetation dynamics. We couple the fire model INFERNO (INteractive Fire and Emission algoRithm for Natural envirOnments) to dynamic vegetation within JULES and use the HYDE (History Database of the Global Environment) land cover dataset to analyse the impact of land-use change on the simulation of present day vegetation. We evaluate the inclusion of land-use and fire disturbance against standard benchmarks. Using the Manhattan Metric, results show improved simulation of vegetation cover across all observed datasets. Overall, disturbance improves the simulation of vegetation cover by 35% compared to Vegetation Continuous Fields (VCF) observations from MODIS and 13% compared to Climate Change Initiative (CCI) from ESA. Biases in grass extent are reduced from -66% to 13%. Total woody cover improves by 55% compared to VCF and 20% compared to CCI from a reduction in forest extent in the tropics, although simulated tree cover is now too sparse in some areas. Explicitly modelling fire and land-use generally decreases tree and shrub cover and increases grasses. The results show that the disturbances provide important contributions to the realistic modelling of vegetation on a global scale, although in some areas fire and land-use together result in too much disturbance. This work provides a substantial contribution towards representing the full complexity and interactions between land-use change and fire that could be used in Earth System Models.

## 1 Introduction

JULES (Joint UK Land Environment Simulator) is a land surface model (LSM) which simulates surface fluxes of water, energy and carbon, along with the state of terrestrial hydrology, vegetation and carbon stores (Clark et al., 2011; Best et al., 2011). It forms the land-surface component in the Met Office Unified Model for Numerical Weather Prediction, as well as in the latest Climate and Earth System Models of the Hadley Centre family including HadGEM3 (Senior et al., 2016) and UKESM1, and can also be used as a stand-alone LSM, used to contribute to international scientific studies such as the Global Carbon Project and TRENDY (Trends in net land atmosphere carbon exchange model intercomparison project). As documented in Cox (2001) and Clark et al. (2011), vegetation cover was previously simulated as a function only of competition between plant species, and a large-scale, spatially-constant disturbance term. Here we document updates to the calculation of vegetation cover, including spatially and temporally varying changes in land-use, and introduce a new disturbance term from fire based on the fire model INFERNO (Mangeon et al., 2016) as separate from the large-scale disturbance factor for the first time in JULES. We use these processes together with dynamic vegetation to address the impact on global vegetation cover.

JULES can be used in a number of different configurations depending on the focus of research, and parameters can be switched on or off by the user accordingly. For example JULES can be used for studying river routing and runoff, snow cover and permafrost, or crop modelling *inter alia*. In this context, it is useful for the community to develop standard configurations that can be used widely, and are thus easily comparable. In this study we use a standard JULES configuration with dynamic vegetation, and focus on the impact of disturbance from fire and land-use on the simulation of vegetation cover.

Land-use change and fire are two of the most important processes which affect vegetation cover. These disturbances affect vegetation dynamics (e.g. Lasslop et al., 2016), atmospheric chemistry (Crutzen et al., 1979), the hydrological cycle (Shakesby and Doerr, 2006) and the carbon cycle (Prentice et al., 2011), as well as surface albedo (López-Saldaña et al., 2015) and feedbacks on radiative forcing. Each year around 4% of vegetation is burnt (Giglio et al., 2013), releasing approximately 2 PgC which equates to around a quarter of emissions from fossil fuel combustion (Hantson et al., 2016; van der Werf et al., 2017). Land-use and land-cover change (LULCC) can include clearance through fire, as well as other forms of deforestation, conversion of natural vegetation to agricultural land, and abandonment of agricultural land with subsequent forest regrowth. At least 50% of the ice-free land surface has been affected by land-use activities over the last 300 years; 25% of global forest area has been lost, and agriculture now accounts for around 30% of the land surface (Hurtt et al., 2011). LULCC can result in changes to biogeochemical and biophysical properties of the Earth system, including changes to surface fluxes of radiation, aerodynamic roughness, heat and moisture, evaporation patterns, soil moisture and latent heat (Betts 2005). LULCC often represents deliberate conversion from one land cover type to another, such as forests to cropland, and this can be long-lasting until the area is subsequently abandoned based on various socio-economic conditions and decision making processes (Turner et al., 1995). Fires may be used in a similar way for land conversion, or otherwise may be unintentional (natural or escaped fire), and thus recovery may be more temporally variable than with LULCC.

LULCC is known to be one of the most important influencing factors in the decline of forests in several ways: directly through deforestation and canopy thinning (cutting as well as use of fire for clearance), and indirectly through fire-leakage which can extend forest losses into much larger areas than planned. Fragmentation is also an important contributing factor, causing increased tree mortality and carbon losses near the forest edges (Laurance et al., 2000), and increased risk of fire spread into the forest (Soares-Filho et al., 2006; Coe et al., 2013; Good et al., 2014). This can be the result of land clearance for agriculture, and for urban expansion. For example there is a clear correlation between distance to roads and increased fire risk in Amazonia (Cardoso et al., 2003). Even when deforestation itself declines, fire incidence can remain high due to increased agricultural frontiers where accidental fires burn out of control (Aragão and Shimabukuro 2010; Cano-Crespo et al., 2015) exacerbated by drought conditions (Aragão et al., 2018). Small-scale forest degradation is sometimes included in the definition of LULCC and can be an important contributor to carbon and biomass loss, however more frequently these contributions are below the level of detection and are often not accounted for in estimates of LULCC (Watson et al., 2000; Arneth et al., 2017). Similarly small fires are difficult to detect by conventional satellite methods (Randerson et al., 2012), leading to potential underestimations in LULCC and emission reporting.

The interaction between fire and managed agricultural land is complex. Small scale croplands are often burnt to clear land before planting or harvesting, and can also be burnt after harvest to dispose of waste, where pasture lands may be burnt to fertilise the soils between crops (Rabin et al., 2017a). Agricultural land may therefore be an important contributing factor in fire emissions, and fire ignition. Conversely, larger agricultural lands may provide a fire break, where more active fire management takes place to prevent fires from spreading into crop areas unintentionally, and it has been shown that burnt area reduces as cropland area increases (Bistinas et al., 2014; ). Andela et al. (2017) has shown that fire occurrence has been reducing in many regions because of agricultural expansion and intensification, making fuel less readily available and decreasing ignitions.

While human ignitions are the main causes of fires in tropical (Cochrane, 2003) and Mediterranean (Mooney et al., 1977) regions, natural fires from lightning and volcanic activity are also important for shaping vegetation cover in temperate (Ogden et al., 1998) and boreal regions (Johnson, 1992; Veraverbeke et al., 2017). In addition, climate-induced land cover change has been shown to be as important in the long-term as anthropogenic LULCC (Davies-Barnard et al., 2015), and can continue to fluctuate for decades before a committed state is realised (Pugh et al., 2018), making it particularly important to incorporate dynamic vegetation processes in modelling (Seo and Kim., 2018). While previous modelling studies have considered the impact of each of these processes (e.g. Sitch et al., 2015; Betts et al., 2015; Seo and Kim, 2018), considering fire, LULCC and dynamic vegetation together is still a relatively recent development.

Future fire activity will depend on a combination of both anthropogenic and climatic factors. Forest susceptibility to fire is projected to change little for low emissions scenarios, but substantially for high emissions scenarios (Settele et al., 2014; Burton et al., 2018). Because the frequency of fires increases with temperature, the IPCC AR5 report concluded that the incidence of fires is expected to rise over the 21st Century (Flato et al., 2013) although there is low agreement in the models on a regional scale due to the complexity of interactions and feedbacks and lack of proper representation in models (Settele et

al., 2014). However while the meteorological conditions may become more conducive to fire risk in the future, the effects of future LULCC will also have a direct impact on how fire risk will change. LULCC can have important impacts on regional climate, and has been shown to reduce evapotranspiration (Cochrane and Laurance 2008), decrease precipitation and induce drought (Bagley et al., 2014), which can in turn initiate abrupt increases in fire-induced tree mortality (Brando et al., 2014;

Castello and Macedo 2016). The interaction of LULCC, climate change and fire is complex (Coe et al., 2013) and in order to understand the multiple feedbacks comprehensively, it is necessary to consider all of these elements together (Aragão et al., 2008). To do this we need to be able to represent these processes explicitly within our models.

Currently the representation of disturbance, in particular fire, drought and tree mortality in models is poorly constrained, as identified in the most recent IPCC report (Ciais et al., 2013; Flato et al., 2013). The purpose of this paper is to document the

developments to JULES to include the explicit representation of fire and LULCC and their coupling to vegetation dynamics, and to evaluate the impact of these developments on the simulation of vegetation within the model, with the aim of ultimately being able to represent these processes within a fully coupled Earth System Model. We begin by describing how dynamic vegetation is simulated in JULES as documented in Cox (2001) and Clark et al. (2011), before describing the new processes of fire and land-use. We then outline the methods used in this study for simulating vegetation cover in a number of experiments,

and describe the benchmarking approach used to quantify the change. We present results showing the impact of fire and LULCC on vegetation cover, which generally decreases woody vegetation cover and increase grass cover, contributing to an improved simulation of vegetation compared to observations.

## 2 Model description and developments

Within JULES a DGVM called TRIFFID (Top-down Representation of Interactive Foliage and Flora Including Dynamics) is

used to represent the carbon cycle and the distribution of different Plant Functional Types (PFTs) (Clark et al., 2011; Cox et al., 2000; Cox, 2001). Here we focus on the simulation of PFT distribution in a global model run. The area of each grid box covered by PFT i: $v_i$ in the original model is determined by species competition and large-scale disturbance:

$$\frac{dv_i}{dt} = \frac{\lambda \Pi v_*}{C_{vi}} \left\{ 1 - \sum_j c_{ij} v_j \right\} - \gamma_v v_*$$

(1)

Equation (1) is used to calculate the evolution of $v_i$. The rate of increase of $v_i$ depends on the carbon available for increasing PFT area ($\lambda \Pi v_*$) and the carbon cost of increasing area, given by the carbon density ($C_{vi}$). Two terms balance the constant expansion of PFTs; a competition term (within the curly brackets) represents the loss of PFT area due to competition for limited space, and a disturbance term ($\gamma_v v_*$) representing vegetation loss due to all mortality processes not related to competition. $\lambda$ is the fraction of NPP per PFT area, $\Pi$, used for increasing PFT area. $v_*$ is a maximum of PFT area, $v_i$, and a minimum of 0.01

gridbox fractional area, imposed to ensure PFTs do not get permanently removed from a given gridbox. $c_{ij}$ determines which

of PFTs i or j is dominant and will out compete the other. $c_{ij}$ is zero for dominant PFTs, meaning the whole gridbox is available for PFT i to expand into; for non-dominant PFTs, $c_{ij}$ is 1 and expansion is scaled by the fraction of the gridbox where PFT i is dominant. The configuration used here has 5 PFTs; the 2 tree PFTs out-compete the shrub PFT, the shrub PFT out-competes the 2 grass PFTs, and the taller of the 2 tree /grass PFTs out-competes the shorter tree /grass PFT. $\gamma_v$ is a PFT-dependent

disturbance rate. The vegetation is updated according to these factors on a 10-day timestep.

Here we include the effects of land-use on vegetation distribution by modifying the competition term of Eq. (1). Similar to competition, land-use is also represented by a limitation to the space available for a PFT to expand into. A fraction of each gridbox is prescribed as the "disturbed fraction", which represents the area covered by agriculture, with no distinction between

cropland and pasture being made. Now adding in land-use to Eq. (1), we have:

$$\frac{dv_i}{dt} = \frac{\lambda \Pi v_*}{C_{vi}} \left\{ 1 - \alpha a_i - \sum_j c_{ij}\, v_j \right\} - \gamma_v\, v_*$$

(2)

Where $\alpha$ is the disturbed fraction and $a_i$ is 1 for non-woody PFTs and 0 for woody PFTs. The three woody PFTs (broadleaf trees, needle-leaf trees and shrubs) are prevented from growing in the disturbed fraction, while the two grass PFTs (C3 grass

and C4 grass) can grow anywhere in the gridbox. Grass PFTs growing in the disturbed fraction are interpreted as agricultural grasses, although they are physiologically identical to "natural" grasses. $\alpha$ can increase or decrease over time. As $\alpha$ increases, first "natural" grasses are relabelled as "agricultural" grasses, then an area of woody PFTs is replaced by bare soil, which can be replaced by the non-woody PFTs over time if they are viable. As $\alpha$ decreases, an area of "agricultural" grasses is relabelled as "natural" and becomes available for woody PFTs to expand into.

**Table 1: The disturbance rate ($\gamma v$) and spreading parameter ($\lambda$) implicitly including fire disturbance (top rows) and excluding fire disturbance (bottom rows).**

| PFT | Broadleaf Tree | Needle-leaf Tree | Shrub | C3 Grass | C4 Grass |
|---|---|---|---|---|---|
| $\gamma_v$ implicit fire | 0.009 | 0.0036 | 0.05 | 0.10 | 0.10 |
| $\lambda$ implicit fire | 3.0 | 3.0 | 1.0 | 1.0 | 1.0 |
| $\gamma_v$ using INFERNO | 0.0045 | 0.0018 | 0.15 | 0.10 | 0.10 |
| $\lambda$ using INFERNO | 1.0 | 1.0 | 1.0 | 1.0 | 1.0 |

The effect of fire on vegetation distribution is included by modifying the disturbance rate, $\gamma_v$. Previously disturbance due to

fire was implicitly included in $\gamma_v$, along with mortality due to pests, windthrow and many other processes. With this new development, fire disturbance, $\beta_i$, is now included as a PFT-dependent burnt area which can vary in space and time. $\beta_i$ is calculated within JULES by the INFERNO (INteractive Fire and Emission algoRithm for Natural envirOnments) fire model

(Mangeon et al., 2016). Now that fire is explicitly represented, $\gamma_v$ must be reduced accordingly, hence the representation of fire does not necessarily increase mortality, but makes it spatially and temporally variable. Table 1 shows the values of $\gamma_v$; in the top row values implicitly include fire disturbance before the coupling, and in the bottom row fire is treated separately using INFERNO. Equation 3, includes fire along with land-use:

$$\frac{dv_i}{dt} = \frac{\lambda \Pi v_*}{C_{vi}} \left\{ 1 - \alpha a_i - \sum_j c_{ij} v_j \right\} - (\gamma_v + \beta_i)\, v_*$$

(3)

The calculation of burnt area depends on fuel availability as documented in Mangeon et al. (2016) and which now includes the additional feedback of reduction in fuel from fire (equation 3). Also included in fuel availability is soil carbon density, providing additional mechanisms by which fire and land-use can feedback onto vegetation distribution. The coupling of fire

and the carbon cycle includes a direct impact of fire on soil; some soil carbon is burnt, resulting in a flux of carbon from the soil to the atmosphere. The burnt soil carbon flux is diagnosed in INFERNO and we now allow the flux to effect the evolution of carbon in the soil pools, $C_k$. The carbon cycle in JULES does not explicitly represent a litter carbon store, however the model includes four soil carbon pools and we use two of these pools as proxies for flammable litter. The decomposable plant material soil carbon pool, $C_{dpm}$, and the resistant plant material soil carbon pool, $C_{rpm}$, both receive the litter carbon flux from

vegetation and have a relatively rapid turnover rates, making them reasonable proxies for the litter carbon store. The calculation of the burnt soil flux is similar to INFERNOs diagnosis of the burnt vegetation flux (equation 8 of Mangeon et al., 2016).

$$f_s = \left( \mu_{min,k} + (\mu_{max,k} - \mu_{min,k})(1 - \theta) \right) C_k \sum_i \beta_i v_i$$

(4)

The efficiency of soil burning is inversely proportional to the saturated soil moisture fraction in the top soil level (0-10cm), $\theta$,

with the values of the completeness of combustion parameters, $\mu$, for each soil pool, k, being based on the original values from INFERNO, and listed in Table 2. The burnt soil flux is proportional to the total available fuel, $C_k$, and the total burnt area, summed over all PFTs.

**Table 2: Completeness of combustion parameters.**

| Soil Carbon Pool | Decomposable plant material, $C_{dpm}$ | Resistant plant material, $C_{rpm}$ |
|---|---|---|
| $\mu_{min}$ | 0.8 | 0.0 |
| $\mu_{max}$ | 1.0 | 0.2 |

Fire and land-use both affect the soil carbon store by altering the vegetation-to-soil litter flux. Without fire or land-use, the litter flux comprises a local litter fall rate, $\Lambda_l$, representing the turnover of leaves, roots and stems, litter due to disturbances and litter due to competition. The total litter fall is defined by Clark et al. (2011) as (their equation 63):

$$\Lambda_c = \sum_i v_i \left( \Lambda_{li} + \gamma_{vi} C_{vi} + \Pi_i \sum_j c_{ij} v_j \right)$$

(5)

Including our new disturbance terms produces:

$$\Lambda_{CvLoss} = \sum_i v_i \left( \Lambda_{li} + (\gamma_{vi} + \beta_i) C_{vi} + \Pi_i \sum_j (\alpha a_i + c_{ij} v_j) \right)$$

(6)

The new term, $\Lambda_{CvLoss}$, still represents a loss of vegetation carbon, but now not all of this flux enters the soil carbon pools, instead some of the vegetation carbon loss due to fire is lost to the atmosphere and some of the loss due to land-use change enters wood product carbon pools. All litter fluxes that do enter soil carbon pools are split between $C_{dpm}$ and $C_{rpm}$ according to PFT-specific parameters as described by Clark et al. (2011). To calculate the losses due to the new processes the vegetation

distribution (Eq. 3) and vegetation loss (Eq. 6) are calculated with and without the new process, and the difference between the two values of $\Lambda_{CvLoss}$ is attributed to the new process.

The litter due to land-use change, $\Lambda_{LUC}$, is calculated by repeating Eq. (3) and (6) with the disturbed fraction from the previous timestep, $\alpha_{-1}$; note that both calculations include some disturbed fraction and it is the litter due to land-use *change* that is being calculated, not the effect of existing land-use.

$$\Lambda_{LUC} = \Lambda_{CvLoss} - \sum_i v_{LUC,i} \left( \Lambda_{li} + (\gamma_{vi} + \beta_i) C_{vi} + \Pi_i \sum_j ((\alpha - \alpha_{-1}) a_i + c_{ij} v_{LUC,j}) \right)$$

(7)

Where, $v_{LUC}$, is the PFT area calculated by Eq. (3) with $\alpha = \alpha_{-1}$. $\Lambda_{LUC}$ is distributed between the soil carbon pools and the wood product pools; the portion that is below ground carbon, given by (root carbon/$C_v$), is added to the soil carbon pools and the remaining above ground portion is added to the wood product pools (Jones et al., 2011).

Carbon loss due to fire, $\Lambda_{Fire}$, is calculated by repeating Eq. (3) and (6) with no burnt area ($\beta = 0$):

$$\Lambda_{Fire} = \Lambda_{CvLoss} - \sum_i v_{NoFire,i} \left( \Lambda_{li} + \gamma_{vi} C_{vi} + \Pi_i \sum_j (\alpha a_i + c_{ij} v_{NoFire,j}) \right)$$

(8)

Where $v_{NoFire}$, is the PFT area calculated using Eq. (3) with $\beta = 0.13$, meaning that 13% of the vegetation carbon killed by fire

is emitted and the remainder enters the soil carbon pools (Li et al., 2012). All terms expressed above are summarised in Table 3.

**Table 3: Summary of terms used in equations 1-8.**

| Variable | Symbol | Unit | Source of variable |
|---|---|---|---|
| Combustion completeness | $\mu$ | | Parameter |
| Competition term | $c_{ij}$ | | TRIFFID |
| Crop indicator | $a_i$ | | Parameter |
| Disturbed fraction | $\alpha$ | Fraction of land surface | Input Map |
| Fire disturbance | $\beta_i$ | $yr^{-1}$ | INFERNO |
| Fraction of NPP allocated to PFT area expansion | $\lambda$ | | Parameter |
| Fractional coverage | $v$ | Fraction of land surface | TRIFFID |
| Large scale disturbance | $\gamma_v$ | $yr^{-1}$ | Parameter |
| Litterfall rate without fire or land-use change | $\Lambda_c$ | kg C m$^{-2}$ yr$^{-1}$ | TRIFFID |
| Local litterfall rate | $\Lambda_l$ | kg C m$^{-2}$ yr$^{-1}$ | TRIFFID |
| NPP per unit of vegetated area | $\Pi$ | kg C m$^{-2}$ yr$^{-1}$ | JULES |
| PFT indices | $i, j$ | | |
| Soil carbon in soil pool k | $C_k$ | kg C m$^{-2}$ | TRIFFID |
| Soil flux | $f_s$ | kg C m$^{-2}$ yr$^{-1}$ | INFERNO |
| Vegetation carbon density | $C_v$ | kg C m$^{-2}$ | TRIFFID |
| Vegetation carbon loss due to fire | $\Lambda_{Fire}$ | kg C m$^{-2}$ yr$^{-1}$ | TRIFFID |
| Vegetation carbon loss due to land-use change | $\Lambda_{LUC}$ | kg C m$^{-2}$ yr$^{-1}$ | TRIFFID |
| Vegetation carbon loss due to litter, fire and land-use change | $\Lambda_{CvLoss}$ | kg C m$^{-2}$ yr$^{-1}$ | TRIFFID |

## 3 Experimental set-up and model evaluation

Here we run JULES Vn4.9 from 1860 to present day with re-gridded CRU-NCEP7 forcing data for climate and $CO_2$, and land-use ancillaries from HYDE 3.2 (History Database of the Global Environment) (Klein Goldewijk et al., 2011), updated to include 2013-2015 as part of the Global Carbon budget, where data were extrapolated based on agricultural trends of the previous 5 years (Le Quéré et al., 2016). The harmonised HYDE dataset estimates fractional agricultural land-use patterns and underlying transitions in land-use annually for 1500-2100, and is spatially gridded at half degree resolution. It does not include impacts of degradation, climate variability, forest management, fire management or pollution on land cover (Hurtt et al., 2006),

and does not specify the nature of the previous land type, whether forested or not (Le Quéré et al., 2016). This was then re-gridded for use in JULES at N96 resolution (1.25° latitude x 1.875° longitude), and implemented from 1860-2015 as annual land-use change as outlined above. Because the process of land-use change excludes woody PFT from agricultural areas, it is expected that there will be a reduction of tree growth and increase of grasses when this term is included.

For the fire experiments, the model was spun-up for 1000 years with fire on using pre-industrial land-use and $CO_2$ at 1860 prescribed as a climatology. INFERNO was run here with constant natural and anthropogenic ignitions, and interactive fire-vegetation on.

The model was tuned with fire towards a PFT distribution from the European Space Agency Climate Change Initiative (ESA CCI, 2010) observations, using maximum spreading (λ) as LAI_min= 1.0, and the large-scale disturbance term ($\gamma_v$) modified

as per Table 1. Altering LAI_min is a way of increasing the rate of spread of vegetation to account for a known deficiency in the model associated with slow regrowth. The large-scale disturbance of trees has been halved and disturbance of shrub increased by a factor of three to be within the error bars of ESA observations.

JULES was configured to the TRENDY set up (Sitch et al., 2015) using two experiments: S2 = $CO_2$ and climate forcing (with land-use constant at 1860, referred to as 'No LULCC, no fire'); and S3 = $CO_2$, climate forcing and land-use change, using the

standard large-scale constant disturbance rate for the purposes of comparison (referred to as 'LULCC only'). These two experiment configurations were then repeated including the new explicit representation of fire for SF2 (referred to as 'fire only') and SF3 (referred to as 'LULCC and fire').

For benchmarking the performance of our model configurations, we use the protocol used by FireMIP (Rabin et al., 2017b) based on the benchmarking system outlined in Kelley et al. (2013). Annual average burnt area was assessed using the

Normalised Mean Error (NME) metric, which sums the difference between the model (mod) and observations (obs) over all cells (i) weighted by cell area ($A_i$) and normalised by the average distance from the mean of observations $\overline{obs}$:

$$NME = \frac{\sum A_i \cdot |mod_i - obs_i|}{\sum A_i \cdot |obs_i - \overline{obs}|}$$

(9)

NME comparisons are conducted in three steps. Step 1 compares simulated and observed annual average burnt area. For step

2, $mod_i$ and $obs_i$ become the difference between modelled or observed and their respective area weighted means, i.e. $x_i \rightarrow x_i - \bar{x}$ , thereby removing systematic bias to describe the performance of the model about the mean. Step 3 additionally removes the mean deviation i.e. $x_i \rightarrow x_i / |\overline{x_i}|$ and describes the models ability to reproduce the spatial pattern in burnt area. Comparisons are made against fire CCI (Alonso-Canas and Chuvieco, 2015), MCD45 (Archibald et al., 2013), GFED4 (Giglio et al., 2013), GFED4s (van der Werf et al., 2017).

Simulated vegetation fractions are compared against Vegetation Continuous Fields (VCF) from MODIS (2002-2012), as recommended for fireMIP analysis (Rabin et al., 2017b), and ESA CCI reference observations, using the Manhattan Metric (MM):

$$MM = \sum_{ij} A_i \cdot |mod_{ij} - obs_{ij}| / \sum_i A_i$$

(10)

Where j is vegetation type. Using the MM, we assess model performance against different vegetation combinations (see table SI-3 for a full list of comparisons).

All benchmark datasets were resampled from their native resolutions to N96 before comparison. Scores for all metrics are directly comparable across models, e.g. a score of 0.6 is twice as close to observations as 1.2, which we describe as 100% improvement from the control as per Kelley et al. (2014). Three null models are used for further interpretation (Table SI-4). The median and mean null model scores compare the median or mean of all observations with the observation data. Randomly-resampled null models compare resampled observations (without replacement) against observations, using 1000 bootstraps to

describe the distribution of the null model. Individual model quality can be described in terms of number of null models exceeded (Table 4).

## 4 Results

Here we present results showing the effect of LULCC and fire on the simulation of vegetation in JULES. First, we present global vegetation by PFT to assess the present day spatial distribution of vegetation as a result of LULCC disturbance compared

to observations. We then move on to fire disturbance, first reviewing how the new fire disturbance term modelled by the coupled INFERNO model compares to GFED observations of burnt area as validation for the fire model. We then present global vegetation by PFT for fire disturbance and show how this compares to observations. Finally we show the global distribution of vegetation in the context of observations considering uncertainty bounds.

Without explicit fire or LULCC disturbance, the model produces too much broadleaf vegetation compared to observations, especially over South America and SE Asia (Fig. 1, second column). Both broadleaf and needleleaf trees are not simulated well in the high latitude boreal regions in JULES, and do not extend far enough across this region, which is not improved by adding disturbance. Overall the model performs poorly at simulating tree cover, as indicated by a MM score of 0.78 for vegetation cover comparison and 0.64 for wood cover (Table 4) when comparing against VCF (generally worse than our null

models – Table SI-4), and 0.72 and 0.45 respectively compared to CCI. The introduction of LULCC generally results in a reduction in broadleaf, needleleaf and shrub vegetation, and an increase in C3 and C4 grasses (Fig.1, fourth column), improving the simulation of vegetation cover by 23% compared to VCF and 17% against CCI (Table 4). This is as expected, with the purpose of this disturbance term being to represent crop area with C3 and C4 grasses. With LULCC, the broadleaf fraction is much improved over South America compared to observations, but is not improved in the high latitude regions. C3 grass is

improved with LULCC, but the fraction is still too low, whereas shrub fraction remains too high (also shown in Fig. 5). The bare soil fraction is too high in the model, but the inclusion of LULCC has little effect on this.

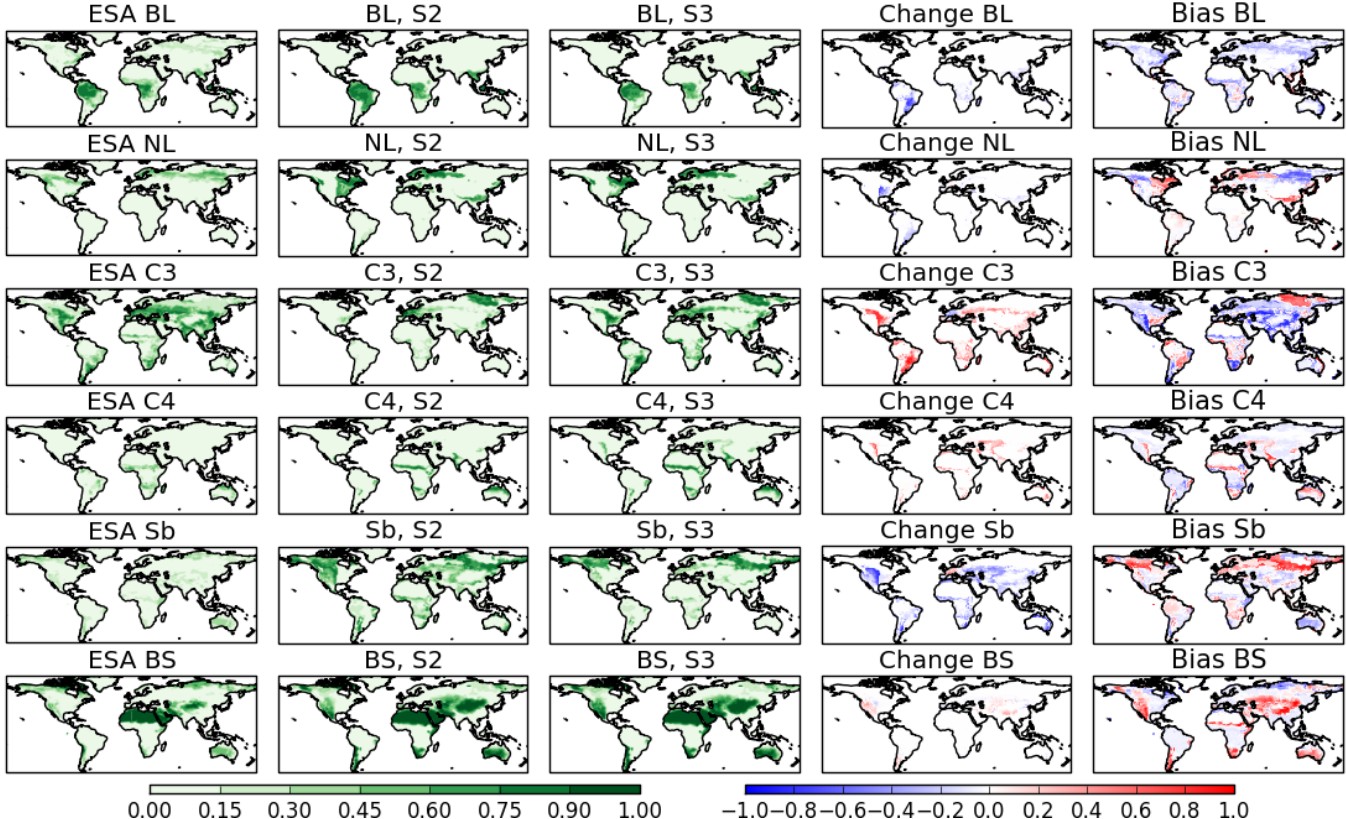

**Figure 1: Present day (2010-2015) vegetation fractions for the TRENDY S2 (no LULCC, no fire) and S3 experiment (LULCC only) by PFT, without fire, compared to observations. Left column shows ESA CCI observations (2010), second column shows vegetation without LULCC (S2), third column shows vegetation with LULCC (S3), fourth column shows the change resulting from LULCC (difference between column 2 and 3), and right column shows bias of S3 compared to observations (difference between column 1 and 3). BL = broadleaf, NL = needleleaf, C3 = C3 grasses, C4 = C4 grasses, Sb = shrub, BS = bare soil.**

Now considering fire, compared to observations of burnt area from GFED 4.1s (including small fires) INFERNO captures the spatial extent and level of fire relatively well (Fig. 2 and Table SI-4), and it is clear that the integrity of the model to accurately simulate global burnt area (as presented in Mangeon et al, 2016) is preserved through the coupling of fire and vegetation, both with and without land-use. INFERNO accurately simulates the areas of high fire occurrence found in GFED4.1s, especially over Africa, northern Australia, South America and SE Asia, although the model also shows high fire occurrence over India which is not seen in the observations. This is likely due the current lack of representation of fire suppression in agricultural and urbanised areas. An NME score of 0.79-0.95 (Table SI-4) outperforms all but one null model, and is better than published assessments of other global fire-vegetation models using the same metrics (Lasslop et al., 2014; Kelley et al., 2013; Kloster & Lasslop 2017; Hantson et al., 2016). NME step 2 and step 3 scores also fall in a similar range (Table SI-4) demonstrating a strong performance in overall fire magnitude, variance and spatial pattern.

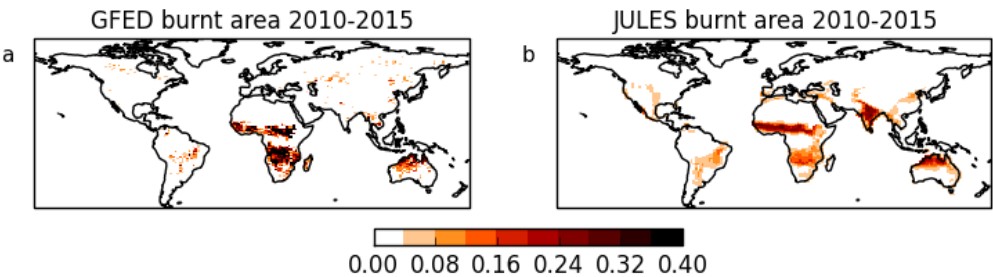

**Figure 2: Average 2010-2015 burnt area from GFED 4.1s observations (a) and as modelled by JULES-INFERNO (b)**

Similarly to LULCC, fire disturbance also improves the representation of vegetation cover, this time by 31% compared to VFC and 11% against CCI (Table 4). The balance of tree to grass cover over South America for example shows particular improvement (Fig. 3, third column), although in other areas fire creates too much disturbance and results in tree fraction being too sparse, notably across Africa (although still within the range of uncertainty, see Fig. 5). C3 grass fractions are generally too low without fire compared to observations, and this is improved with fire. C4 grasses are well modelled both with and

without fire (Table SI-4). The shrub fraction is too high in the model compared to observations, but this is also improved when fire is included (28%, Table SI-4, and also shown in Fig. 5). There is too much bare soil in the model without disturbance, and this increases further with fire. The overall change as a result of fire is generally a reduction in the larger PFTs (broadleaf and needleleaf trees) and an increase in grasses and bare soil (Fig. 3, fourth column). Broadleaf trees show a loss in all regions, including the Cerrado region to the south of the Amazon, across the arid regions in Africa, SE Asia, and northern high latitudes.

The changes in shrub and C4 grasses are more variable, and are region-dependent. The increase in grasses and bare soil reflects the burnt area as modelled by INFERNO (Fig. 2b), indicating shift away from woody vegetation (broadleaf trees, needleleaf trees and shrubs) towards faster growing vegetation and bare ground as a result of fire.

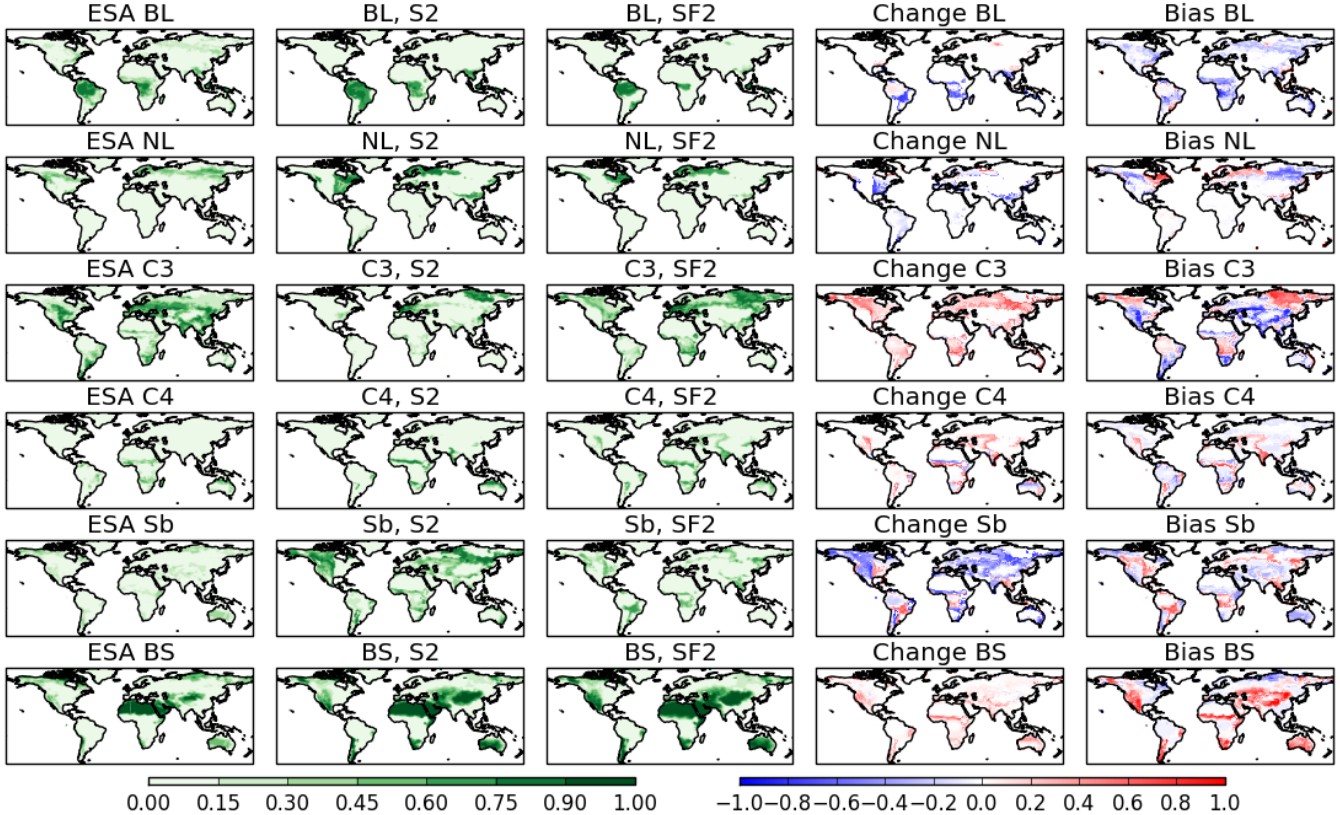

**Figure 3: Present day (2010-2015) vegetation fractions for the TRENDY S2 experiment (no LULCC, no fire) and SF2 (fire only) by PFT compared to observations. Left column shows ESA CCI observations (2010), second column shows vegetation without fire or LULCC (S2), third column shows vegetation with fire only (SF2), fourth column shows the change resulting from fire (difference between column 2 and 3), and right column shows the bias of SF2 compared to observations (difference between column 1 and 3). BL = broadleaf, NL = needleleaf, C3 = C3 grasses, C4 = C4 grasses, Sb = shrub, BS = bare soil.**

As with all observational datasets, there are uncertainties associated with retrieving observations of land cover and the classification of these into a small number of plant functional types. The observations used here are from ESA CCI, which have been processed into the 5 PFTs used by JULES so as to be comparable with the model output (Hartley et al., 2017), introducing a range of possible values for each vegetation type. The representation of vegetation distribution is further complicated by the seasonal variation, where peak growing season will have higher fraction of vegetation than low season, and high fire-risk areas will show burnt area as high bare soil in peak fire season. These uncertainties give a range of potential vegetation cover, and the developments to the representation of disturbance in JULES described here have been tuned to give reasonable distribution within this range of uncertainty as far as possible (Fig. 4 and Fig. 5 top left panel). The 'best estimate' of vegetation cover from ESA, known as the reference case, is otherwise used for comparison, and VCF used to provide additional comparison in the benchmarking assessment.

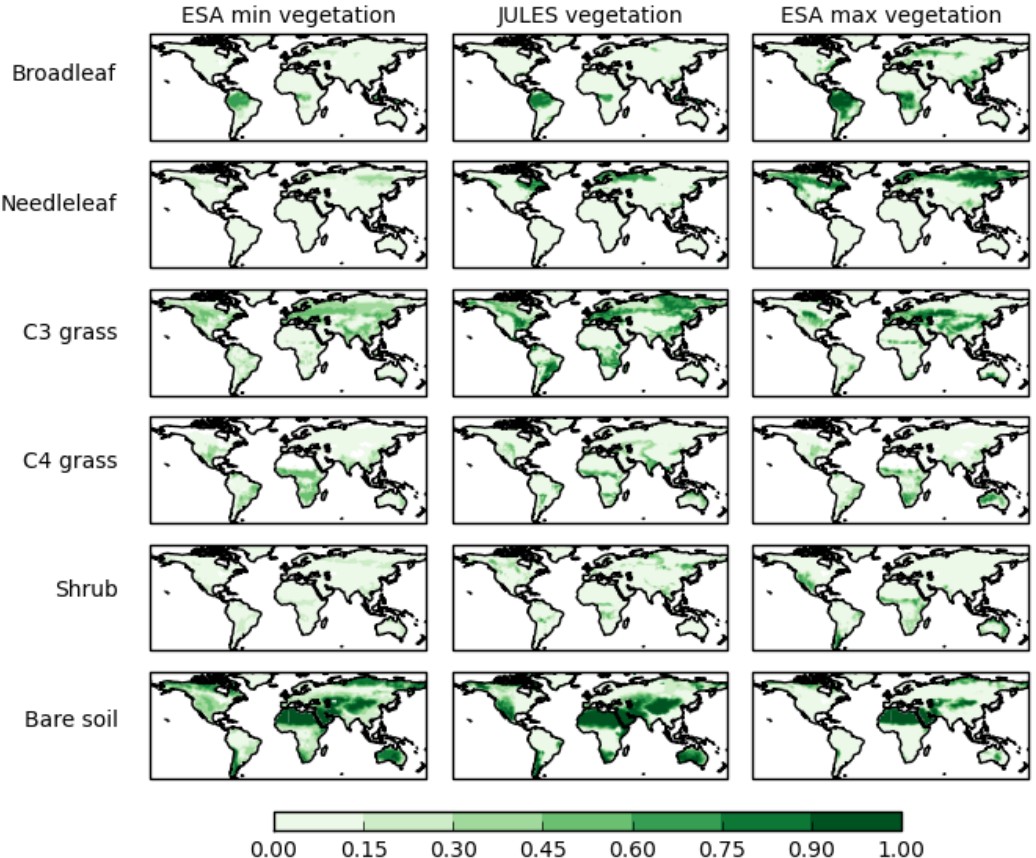

**Figure 4: Present day (2010-2015) vegetation fraction PFT, as modelled by JULES with LULCC and fire (SF3, central column), compared to the range of uncertainty from ESA CCI observations (V1, 2010) (Minimum fractions left column, maximum fractions right column). Top row = broadleaf, second row = needleleaf, third row = C3 grass, fourth row = C4 grass, fifth row = shrub, sixth row = bare soil.**

Considering the distribution by vegetation type (trees, grasses, shrubs and soil), in all cases adding disturbance to the model brings the global total vegetation closer to reference observations, although bare soil increases in the opposite trend (Fig. 5, top left panel). In the case of trees and shrubs, fire plus LULCC creates too much disturbance (42% and 47% less coverage than observations respectively), but grasses increase (13% more coverage than observations) (Table SI-1). This is reflected in only slight improvements in MM scores for vegetation cover and wood cover comparisons against VCF, and a slight degradation when compared against CCI (Table 4). Trees are reduced by 43% when both disturbances are included (S2 compared to SF3), shrubs by 71%, and grasses increase by 127% (Table SI-1), taking into account the updated terms for $\gamma_v$ (Table 1). There is an increase of 20% in bare soil with disturbance included. Overall, adding disturbance into JULES reduces

the bias of shrubs from 72% to -47%, and grasses from -66% to 13% compared to observations (Table SI-1). All results show statistically significant difference with disturbance compared to no disturbance (Table SI-5).

However there is more variation by biome. In all cases tree fraction is simulated as too low with both fire and LULCC although the extent of this varies (Fig. 5). In some cases shrubs improve (in the temperate and boreal forests), but in others the results show too much disturbance (tropics, savanna and temperate grasses). Grasses are generally higher than observations, except for the temperate grasses biome. Both disturbance terms reduce the tree and shrub fractions, and increase grasses and bare soil fractions. In most biomes bare soil fraction is too high compared to observations, except in the tropics and boreal regions where the fraction is well represented compared to observations.

Overall, the inclusion of these disturbance terms within JULES leads to a shift towards grass cover and a reduction in woody PFTs. This is as expected for land use, which replaces trees with grasses as a representation of crops. The regrowth rates for trees is much slower than for grasses, which spread fast and recover quickly (see section 2), which may be an important factor in the response to fire. With continuous disturbance which varies spatially and temporally now included in the model, the vegetation seems unable to recover trees in some areas, notably around the Cerrado and Congo regions, instead encouraging the growth of grasses in their place (Fig. 4).

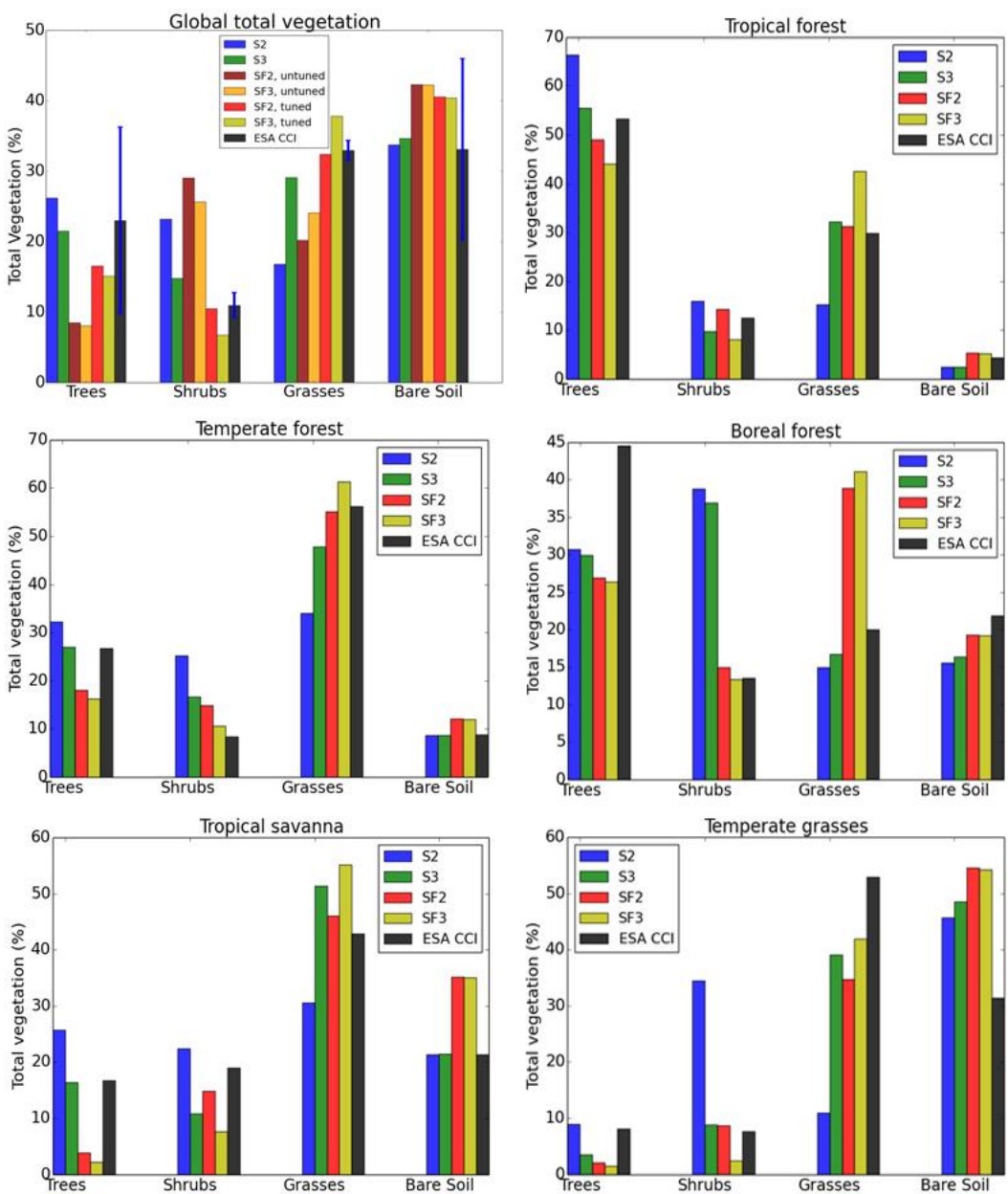

**Figure 5: Present day (2010-2015) total vegetation cover (percentage) globally (top left), and by WWF biome (5 out of 8 shown here: tropical forest, temperate forest, boreal forest, tropical savanna, and temperate grasses. Tundra, Mediterranean wood and desert not shown). Trees = total broadleaf and needleleaf trees, grasses = total C3 and C4 grasses. Top left panel includes results prior to tuning, plus uncertainty bars for the ESA observations of vegetation cover shown in blue.**

Table 4: Benchmarking results for each experiment by vegetation type, using VCF and CCI reference observations. Vegetation cover = woody, grass and bare soil cover for VCF and tree, shrub, grass and bare soil cover for cci. Woody cover = trees and shrubs vs other cover. Trees = BL and NL vs other cover. Grass cover = grass vs non-grass cover. Lower results for JULES = closer to observations. Colours indicate how many null models the configuration exceeds: Blue = all; green = all but one; yellow = only exceeds one; red = none exceeded.

| Comparison | Observations | JULES | | | | Improvement from control | | |
|---|---|---|---|---|---|---|---|---|
| | | S2: Control | S3: Land use only | SF2: fire only | SF3: Land use and fire | S3 | SF2 | SF3 |
| Vegetation cover | VCF | 0.78 | 0.60 | 0.54 | 0.51 | 23.08% | 30.77% | **34.62%** |
| | CCI | 0.72 | 0.60 | 0.64 | 0.63 | 16.67% | 11.11% | **12.50%** |
| Tree Cover | CCI | 0.35 | 0.28 | 0.30 | 0.30 | 20.00% | 14.29% | **14.29%** |
| Wood Cover | VCF | 0.64 | 0.43 | 0.33 | 0.29 | 32.81% | 48.44% | **54.69%** |
| | CCI | 0.45 | 0.31 | 0.35 | 0.36 | 31.11% | 22.22% | **20.00%** |
| Grass Cover | VCF | 0.64 | 0.48 | 0.43 | 0.42 | 25.00% | 32.81% | **34.38%** |
| | CCI | 0.43 | 0.33 | 0.40 | 0.42 | 23.26% | 6.98% | **2.33%** |

## 5 Discussion

### Impact of fire and land-use changes

Fire and land-use are important global disturbances, and the results presented here have shown that when considered, they have a significant impact on the modelled vegetation as represented by JULES. In all cases, including disturbance brings the vegetation fractions closer to the observations compared to no disturbance, although in some cases there is a tendency towards too much disturbance when both fire and LULCC are included, and bare soil increases too much compared to observations (Fig. 5). Disturbance generally improves the simulation of shrubs and grasses, but tree fractions are often simulated as too sparse. LULCC mainly decreases trees and shrubs and replaces them with C3 and C4 grasses (representing crop and pasture). Fire creates a more mixed response, decreasing vegetation in the boreal regions and high fire risk areas, and showing an increase in grasses. Both fire and LULCC reduce the larger vegetation types when added to the model (Fig. 1 and 3). Without the inclusion of fire, this could result in an over-estimation in the amount of carbon released due solely to LULCC, which may have significant impact on carbon budgets.

Previous work has shown that fire may be an important contributor to the existence of savannas (Cardoso et al., 2008; Bond et al., 2005; Staver et al., 2011). The results shown here seem to support this conclusion, showing that when fire is included in the model there is a shift towards open savanna-like states in areas that climatologically could support trees without the incidence of fire, including the Cerrado area of South Brazil, and savanna areas in Africa. Here we have shown that a large savanna region in South America is completely forested in the model without the addition of fire or anthropogenic LULCC.

**Uncertainty**

Here we have used the ESA CCI land cover product as our observational data for comparison with the model output. The CCI product has been translated into the 5 PFTs that are used in JULES (Poulter et al., 2015), and through the process of data collection and classification, a number of uncertainties are introduced which result in a range of possible outcomes for land cover distribution (Hartley et al., 2017). These uncertainties can include variation in classifying the surface reflectance products into the 22 land cover classes, and aggregating these by dominant vegetation type into just 5 PFTs for JULES using a consultative cross-walking technique. This classification also takes into account seasonal variation in NDVI (greenness), burnt area, cloud cover and snow occurrence that can all vary throughout the year, giving a large range between the minimum and maximum possible vegetation cover for any one PFT, as shown in Fig. 4 and the blue bars in Fig. 5. For this reason we also use the MODIS VCF for benchmarking comparison. The VCF product is a characterisation of the land surface into just three components of ground cover using satellite data: tree cover, non-tree vegetation cover, and bare ground. The model performs well compared to a simple classification of tree and non-tree vegetation cover, showing the spatial coverage of vegetation is simulated well when both disturbances are added to JULES. The benchmarking results compared to CCI still show an improvement compared to the control, but on a global scale this is better when each disturbance is considered separately, suggesting further parameterisation may be beneficial for each PFT. However, it is important to consider regional improvements or degradation as well which can be masked in global scale analyses (Figs. 1,3,5). It also suggests that there may be some overlap in the disturbances, which reflects the complicated nature of how fire and LULCC are often used together for land clearance, and future development would benefit from reducing burnt area in cropland areas (Bistinas et al., 2014). The HYDE LULCC dataset in this study has been developed from a combination of model, satellite and historical reconstructions of agricultural and population data, and the biomass quantities are noted to contain uncertainties due to lack of direct observations from the historical period (Hurtt et al., 2011). Some of what has been attributed to LULCC may include fire clearance, which is a key point for consideration for other DGVMs including fire and land use together.

**Limitations and future developments**

When interactive fire was initially added to JULES, there was a tendency towards complete dominance by shrubs and significant tree reduction (see Fig. 5, top left panel). This was tuned to the observations by increasing the large-scale disturbance term ($\gamma$) and increasing spreading ($\lambda$) (Table 1), to account for the fact that fire was previously included in the total mortality rate. Grasses spread and recover quickly with TRIFFID, whereas larger PFTs take longer to re-establish. On this timescale the tree cover is not able to recover fast enough with constant disturbance from fire, and the results indicate that fire restricts tree growth and encourages a shift towards the more responsive vegetation types. There are a number of ways this could be addressed in future developments. Grasses can be given a higher mortality rate to prevent over-growth, but this has been tested and results in too much bare soil because trees are unable to recover. The fractions were low from the start of the run (1860) as fire was included in the spin-up, and the vegetation does not recover through the transient simulation due to continual disturbance, leading to present day levels being low. This perhaps points to a need to develop faster regrowth of trees

within TRIFFID to cope with disturbance, for example by representing age or mass classes within each PFT to enable a range of successional stages to be represented. It is also worth noting that the fire disturbance is high in some areas in the model compared to observations (Fig. 2), which may lead to too much disturbance in these regions, whereas in other areas the burnt area maximum is underestimated. In addition, there remains significant underlying complexity around the interaction of

LULCC and fire as discussed in section 2. For example, agricultural land in some regions may be a cause of fire ignition, whereas in other areas may act as a fire break or generate anthropogenic fire suppression, and future development would benefit from reducing burnt area in cropland areas (Bistinas et al., 2014). One way forwards for this could be to identify the average field size based on surrounding vegetation, and mask fire in larger agricultural regions, but allow smaller fields to include the probability of burning, or include fragmentation effects such as described in Pfeiffer et al. (2013). There will also

be additional complexity around the PFTs themselves where some species will be more fire resilient than other species, for example vegetation in high fire-risk areas often develops thicker bark for protection from fire, whereas other species may adapt to the fire and use it as a method of reproduction and resprouting (Kelley et al., 2014; Pellegrini et al., 2017). The representation of just five PFTs is a considerable simplification of the real world. Finally, we have just considered two of the main disturbances here. We have not considered windthrow, pests, and diseases etc., which for now are still aggregated into the

generic large-scale disturbance term in JULES.

There are still a number of regions that require improvement in the simulation of vegetation. In all of the JULES simulations there are too few needleleaf trees across the boreal regions compared to observations. With fire, notably the trees across the extratropics and savanna regions such as the Congo region in Africa is reduced. Further work could be to develop these configurations into the 9 PFT set up by (Harper et al., 2016). In particular, recent work has shown that the distinction between

evergreen and deciduous needleleaf trees has led to an improved representation of boreal forests within JULES which could improve these simulations (Harper et al., 2018).

## 6 Conclusion

This work has described the first steps in developing the land surface model JULES to represent fire and land-use as separate disturbances. The results have shown the significant contributions of these disturbances to changes in vegetation on a global

scale. Without disturbance JULES simulates too much vegetation in most PFTs compared to observations, which is generally improved with the addition of fire and LULCC, although there is still regional variation. Disturbance generally has the effect of decreasing tree cover (43%) and shrubs (71%) and increasing grasses (127%). In places the disturbance is too high with both fire and LULCC and leads to vegetation being reduced too much. Slow regrowth rates in TRIFFID also mean that with constant disturbance from fire, there is a shift towards faster growing PFTs that can recover and spread quickly. Overall,

representing disturbance in JULES improves the simulation of total vegetation cover compared to both VCF and CCI datasets, by 35% and 13% respectively, with woody vegetation improving by 55% and 20% respectively. The simulation of shrubs and grasses is much improved, with the bias reducing from 72% to -47%, and from -66% to 13% respectively. It is expected that

fire risk will increase in the future with climate change as a result of hotter, drier conditions, but fire occurrence depends heavily on the interaction with LULCC. The developments to the model that have been outlined in this paper now give the capability to model future interactions between fire and LULCC and the impact that this could have on future vegetation density, spread and carbon storage. Overall we have presented results for an improved representation of mechanistic processes

of disturbance in JULES using a non-optimised approach, with positive results to vegetation cover. This is a significant first step in the representation of highly complex factors surrounding anthropogenic and natural disturbances in the model, and lays the foundation for future developments into Earth System Models.

**Code Availability**

The JULES code used in these experiments is freely available on the JULES trunk from version 4.8 (revision 6925) onwards.

The rose suite used for these experiments is u-ap845, at Vn4.9 r9986. Both the suite and the JULES code are available on the JULES FCM repository: https://code.metoffice.gov.uk/trac/jules (registration required).

**Acknowledgements**

This work and its contributors (CB, RB, CJ, ER, AW) were supported by the Newton Fund through the Met Office Climate Science for Service Partnership Brazil (CSSP Brazil).

MC acknowledges support from the Brazilian BNDES/Amazon Fund Project MSA/BNDES/BIOMASSA-SUB 7.

The contribution by DK was supported by the UK Natural Environment Research Council through The UK Earth System Modelling Project (UKESM, Grant No. NE/N017951/1)

We would like to thank Nicolas Viovy and Philippe Ciais for making available their CRU-NCEP forcing data, and for their kind permission for its use in these model runs.

**Author Contribution**

CB updated the code in the JULES trunk to include fire mortality, with help and advice from ER, AW, RB, CJ, AH. CB drafted the text and made the figures. ER and CB co-wrote section 3 and the equations. DK performed the benchmarking. All authors have contributed to the analysis methods and to the text. The authors declare that they have no conflict of interest.

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
