# Peer review of "Representation of fire, land-use change and vegetation dynamics in the Joint UK Land Environment Simulator Vn4.9 (JULES)"

_Geoscientific Model Development, 2018_

## Referee Comment (RC1) · Anonymous Referee #1 · 20 Sep 2018

In this work the authors attempted to evaluate the representation of the land-use change and fire as separate disturbances on the simulated vegetation covers in a land surface model (i.e., JULES). The structure of the paper is loose while the context and figure quality may need an improvement. Some issues with respect to the method descriptions is not accurate and sounds vague. The clarification of these issues is critical to understand results presented in this study. I recommend the major revision of the paper before the possible acceptance of GMD by addressing my following comments.

Major comments:

1.  The section 2 reads like a literature review on the interaction between fire and

[Figure]

LULCC and is suggested to be included in the Introduction.

2. In the abstract and the context of the paper, the authors used a lot of "up to xx%". I don't think this quantification metric is sound because it stands for the maximum situation. Please use median or mean for the quantification.

3. In the paper, the authors used the HYDE data to represent the land-use change. However, the citation of this dataset is not accurate. Please include the original publication of this dataset to appreciate the efforts by the dataset developers. Also, please add the dataset version used in this work and longer description of this dataset. http://themasites.pbl.nl/tridion/en/themasites/hyde/publications/index-2.html

4. It is not clear to me that the unit of each variables in Eqns. (1)-(8) in the paper. Could you clarify the unit of each variable in the revision and make sure the the unit is consistent between the left-hand and right-hand of equations?

5. Page 6, lines 16-17, the authors calculated the litter due to land-use change from the previous time step. What is the time step of the model? Since the land-use change is yearly data, how do you incorporate the land-use change data in the model?

6. Page 7 line 4, the authors noted the model version of JULES as Vn4.9 but the model version in the title of the paper is Vn4.8. Please correct one of them to be consistent.

7. Page 7 Line 18, what does the TRENDY stand for?

8. The authors emphasized that they made an attempt to improve JULES by including EXPLICIT representation of fires and land-use change. Also they mention "Previously in JULES, fire disturbance has not been represented as a separate process, but included in a generic large-scale disturbance term as a spatially-constant turnover rate" (in Page 2, lines12-14). According to Eqns., the fire disturbance in this work is PFT dependent. My question is: What is the difference in the impact of fire disturbances on vegetation covers between the explicit PFT-dependent treatment of fire (implemented in this work) and the previous simple treatment with constant disturbance from fires? If

you run a new simulation S4, the difference between S4 and SF2 should be able to tell you if there is any improvement of this explicit treatment of fire or not compared with the previous treatment of constant disturbance. Does this explicit treatment of fire disturbance improve vegetation representations through all vegetation types or just within specific vegetation types?

9. According to Table SI-4, the burned area and seasonal phase simulated in this work does not have so much difference between S2F and S3F. By visual comparison, I did not see much difference in burned area between S2F and that present in in Figure 2 of Mangeon et al. (2016). You may state this with respect to burned area in the context according to Table SI-4 and Figure 2.

10. In Figure 5, what does uncertainty bar stand for? Does that relate the spatial uncertainty? Please clarify.

11. According to the figure given in the last column of Table 3 (i.e., improvement from control), I figured, for instance for S3, the improvement (%) = |S3-S2|/S3x100 (i.e., |0.6-0.78|/0.6=0.3). Should the percentage improvement be |S3-S2|/S2x100 since S2 is the control simulation? Please clarify this metric in the method section. Also, please calculate statistical significance regarding to this improvement?

12. The color bar of Figure 3 partially appears. Please fix it.

13. The font size of figure labels is not consistent (comparing Fig. 4 vs Fig. 5). Please fix it.

References: Mangeon, S., Voulgarakis, A., Gilham, R., Harper, A., Sitch, S., and Folberth, G.: INFERNO: a fire and emissions scheme for the UK Met Office's Unified Model, Geosci. Model Dev., 9, 2685-2700, https://doi.org/10.5194/gmd-9-2685-2016, 2016.

---

## Referee Comment (RC2) · Anonymous Referee #2 · 10 Oct 2018

The methods presented in the Burton et al. manuscript refer to the implementation of land-use and fire as a disturbance effect in the JULES land model. The manuscript does not present a clear modelling concept of how the both processes are presented, in terms of qualitative description and/or supporting it by a flow-chart which would also guide the reader through the manuscript. The fact that land-use can be regarded as a disturbance is flawed because land-use change is a permanent, very often irreversible change in land-cover. It is reversible when people abandon their fields and that depends on socio-economic conditions that motivate human decision-making. Such a reflection is missing in the introduction. Introduction The literature overview leaves the reader with an unclear message, other than it is very complex. However, the literature,

also the cited does allow to conclude which processes are essential to incorporate land-use and fire disturbance in land surface models such as JULES. The problem statement that DGVMs have to properly consider disturbances has been identified already in papers in, e.g. Foster et al. 1998, and has been implemented in many ways in many DGVMs since then. This applies also to land-use.

The methods section starts with explaining how the disturbance term is implemented in the major equation on quantifying changes in vegetation. And here starts the problem of the modelling approach: what is presented is a simply cookie-cutter approach to correct PFT coverage by the proportion of fire disturbance and land-use. Such an approach represents the level of science of the 1990ies. Since then many more advanced approaches also simple ones have been published from which this modelling concept can profit. The remainder introduction of equations in the methods section is referring to already published modelling studies and the text does not explain how this was adapted to the current model version or what was updated given the latest progress in science in that field. Therefore, I cannot identify any added scientific value in terms of modelling approaches from which other modelling groups would profit. Variables in equation 1 are insufficiently defined or explained. The feedback of fuel availability on vegetation distribution is not explained (equation 3). From this starting point or poor modelling concept and inadequate description, it makes in my view no sense to review the remaining part of the manuscript because it makes it impossible to judge if the results produced are based on solid ground or if they can be reproduced. Unfortunately, I cannot give a more positive feedback on this manuscript. But I hope a deeply revised version of the modelling approach will allow to do so in the future.

---

## Author Comment (AC1) · 19 Oct 2018

**Response to Reviewer 1**

*Anonymous Referee #1*

*Major comments:*

*1. The section 2 reads like a literature review on the interaction between fire and LULCC and is suggested to be included in the Introduction.*

This section was intended to provide some context to the processes of fire and LULCC, so we are happy to include this section in the Introduction. We will rearrange section 1 and 2 to accommodate this change.

*2.  In the abstract and the context of the paper, the authors used a lot of "up to xx%". I don't think this quantification metric is sound because it stands for the maximum situation. Please use median or mean for the quantification*

The comparison we use in the paper is between two different observational datasets, so we don't believe a mean is valid in this case. Instead we will amend the text so that both numbers are referred to rather than using "up to xx%", and hope this satisfies the point.

*3.  In the paper, the authors used the HYDE data to represent the land-use change. However, the citation of this dataset is not accurate. Please include the original publication of this dataset to appreciate the efforts by the dataset developers.  Also, please add the dataset version used in this work and longer description of this dataset.*
*http://themasites.pbl.nl/tridion/en/themasites/hyde/publications/index-2.html*

Apologies that this reference was omitted in the manuscript. We will include the original Klein Goldewijk (2011) reference, and add further information on the dataset.

*Klein Goldewijk, K. , A. Beusen, M. de Vos and G. van Drecht (2011). The HYDE 3.1 spatially explicit database of human induced land use change over the past 12,000 years, Global Ecology and Biogeography 20(1): 73-86.DOI: 10.1111/j.1466-8238.2010.00587.x.*

*4.  It is not clear to me that the unit of each variables in Eqns.  (1)-(8) in the paper. Could you clarify the unit of each variable in the revision and make sure the the unit is consistent between the left-hand and right-hand of equations?*

We thank the reviewers for making this important point. We will include an additional table in the 'Model description and developments' section to give all the units used in the equations.

*5.  Page 6, lines 16-17, the authors calculated the litter due to land-use change from the previous time step. What is the time step of the model? Since the land-use change is yearly data, how do you incorporate the land-use change data in the model?*

The agricultural land fraction is altered annually, but the dynamic vegetation updates on a 10 day-timestep. Thus the grasses within the agricultural fraction, and all vegetation outside of the agricultural fraction, update on this timestep according to competition and disturbance as described in the paper. We will include additional text in the 'Model description and developments' section explaining that the vegetation is updated on a 10-day timestep to clarify this.

**6. Page 7 line 4, the authors noted the model version of JULES as Vn4.9 but the model version in the title of the paper is Vn4.8. Please correct one of them to be consistent.**

Apologies for the confusion in the version numbers. The title refers to the version of code where these changes were implemented (Vn4.8), but the model set up we used for this analysis was the most up-to-date version available at the time, which was then Vn4.9. We will amend the title to Vn4.9 for consistency.

**7. Page 7 Line 18, what does the TRENDY stand for?**

TRENDY is not a direct acronym, but refers to a carbon cycle model intercomparison project exploring 'Trends in net land atmosphere carbon exchanges', as outlined on a number of project websites:
http://www.globalcarbonatlas.org/en/content/land-models
http://dgvm.ceh.ac.uk/

and referred to in a number of papers including:

Liu, Z., Ballantyne, A., Poulter, B., Anderegg, W., Li, W., Bastos, A., and Ciais, P. (2018) Precipitation thresholds regulate net carbon echange at the continental scale. Nature communications, 9, 3596 DOI https://doi.org/10.1038/s41467-018-05948-1

Matthew Cervarich, Shijie Shu, Atul Jain, Almut Arneth, Josep Canadell, Pierre Friedlingstein, Richard A Houghton, Etsushi Kato, Charles Koven, Prabir Patra, Ben Poulter, Stephen Sitch, Beni Stocker, Nicolas Viovy, Andy Wiltshire, Ning Zeng (2016) The terrestrial carbon budget of South and Southeast Asia. Environmental Research Letters 11, 105006, 19 October 2016. DOI:10.1088/1748-9326/11/10/105006.

We will add a short definition to the text to clarify.

**8. The authors emphasized that they made an attempt to improve JULES by including EXPLICIT representation of fires and land-use change. Also they mention "Previously in JULES, fire disturbance has not been represented as a separate process, but in-cluded in a generic large-scale disturbance term as a spatially-constant turnover rate" (in Page 2, lines12-14). According to Eqns., the fire disturbance in this work is PFT dependent.**
**My question is: What is the difference in the impact of fire disturbances on vegetation covers between the explicit PFT-dependent treatment of fire (implemented in this work) and the previous simple treatment with constant disturbance from fires? If you run a new simulation S4, the difference between S4 and SF2 should be able to tell you if there is any improvement of this explicit treatment of fire or not compared with the previous treatment of constant disturbance. Does this explicit treatment of fire dis-turbance improve vegetation representations through all vegetation types or just within specific vegetation types?**

The analysis throughout the paper discusses the impact of adding in explicit fire disturbance in SF2 (fire only) and SF3 (fire with land-use) compared to the standard constant disturbance rate in S2 (no fire, no land-use) and S3 (land-use only), as shown by figure 3, figure 5 (blue vs red bars for fire-only disturbance), and SF2 in Table 3 (fire-only disturbance). We acknowledge that this was not described adequately in the 'Methods' section, so we will update the description to make this clearer as follows:

"JULES was configured to the TRENDY set up (Sitch et al., 2015) using two experiments: S2 = $CO_2$ and climate forcing (with land-use constant at 1860, referred to as 'No LULCC, no fire'); and S3 = $CO_2$, climate and land-use forcing, using the standard constant disturbance rate for the purposes of comparison (referred to as 'land-use only'). These two experiment configurations were then repeated including the new explicit representation of fire for SF2 (referred to as 'fire only') and SF3 (referred to as 'land-use and fire')."

***9. According to Table SI-4, the burned area and seasonal phase simulated in this work does not have so much difference between S2F and S3F. By visual comparison, I did not see much difference in burned area between S2F and that present in in Figure 2 of Mangeon et al. (2016). You may state this with respect to burned area in the context according to Table SI-4 and Figure 2.***

We thank the reviewer for highlighting this important point. We will include the following text in the third paragraph in the 'Results' section to makes it clear that the coupling as described in this paper does not degrade the capability of the model to simulate burnt area as previously presented in Mangeon et al (2016): "it is clear that the integrity of the model to accurately simulate global burnt area (as presented in Mangeon et al, 2016) is preserved through the coupling of fire and vegetation, both with and without land-use". Further, this the first time INFERNO has been assessed using the same NME metrics that have been used to assess the capability of other fire models, which we believe is an important part of this analysis to show how well the model performs in the simulation of burnt area.

***10. In Figure 5, what does uncertainty bar stand for? Does that relate the spatial uncertainty? Please clarify.***

Yes, this relates to spatial uncertainty in the extent of the PFT cover, as described and shown in figure 4 and again in the Discussion section. We will amend the caption for figure 5 to make this clearer.

***11. According to the figure given in the last column of Table 3 (i.e., improvement from control), I figured, for instance for S3, the improvement (%) = |S3-S2|/S3x100 (i.e., |0.6-0.78|/0.6=0.3). Should the percentage improvement be |S3-S2|/S2x100 since S2 is the control simulation? Please clarify this metric in the method section. Also, please calculate statistical significance regarding to this improvement?***

We used |S3-S2|/S3x100 in the paper following the published methodology in Kelley et al (2014) as recommended by reviewers for that paper (see published reviewer comments on GMDD https://www.geosci-model-dev.net/7/2411/2014/gmd-7-2411-2014-discussion.html ). However we agree that |S3-S2|/S2x100 is the more common method of calculating the percentage improvement, and so will amend the methodology for this paper. We will look into appropriate methods for testing the statistical significance.

***12. The color bar of Figure 3 partially appears. Please fix it.***

We will amend Figure 3 to fix the colour bar.

**13. The font size of figure labels is not consistent (comparing Fig. 4 vs Fig. 5). Please fix it.**

We will amend the font sizes in all the figures to be consistent.

---

## Author Comment (AC2) · 19 Oct 2018

**Response to Reviewer 2**

*Anonymous Referee #2*

We thank the reviewer for taking the time to share their opinions on our work. However, we wholly disagree with reviewer 2 regarding the novelty and quality of our work presented here. It is factually incorrect to say that what we present was common place in the 1990s in the context of global land surface modelling.

A chronology of model complexity, as used in major international studies and IPCC reports clearly demonstrates this:

- The first ever inter comparison of multiple coupled climate-carbon cycle models, "C4MIP", was published in 2006. Friedlingstein et al (2006; J. Climate). Of the 7 full GCMs included, only 2 had dynamic vegetation enabled and neither included land-use change or fire.
- In 2013 the most recent IPCC Assessment report, AR5, carbon cycle chapter assessed results from 10-15 coupled climate-carbon cycle Earth System Models (Ciais et al., 2013; WG1 Chapter 6). 4 models included dynamic vegetation and land-use change, but only 2 of these included fire (See table 6.11). Both WG1 and WG2 Assessment reports identify fire as a key process which is not routinely included in future projections.
- Land surface models being run offline often have higher degrees of complexity and process inclusion, for example within the "TRENDY" set of models used each year for the Global Carbon Budget update (Le Quere et al., 2017) some now include dynamic vegetation, land-use and some form of fire, but few of these processes are routinely coupled into a GCM yet as outlined above. This is therefore still an active area of research.

JULES is central to major international scientific studies such as the Global Carbon Project and the "TRENDY" project, and forms the land-surface component of HadGEM3 and UKESM1. It is therefore clear that developing JULES to build on the existing vegetation dynamics and land-use representation is both vital and cutting edge. Such a model will contribute significantly to IPCC AR6 and future Global Carbon Budget updates, as well as significantly advancing this community land surface modelling capability available to a wide user base. Documenting this clearly in the literature and making this publicly available as a matter of transparency and as a basis for further developments is an essential element of this, which is what our paper attempts.

Furthermore, we would be interested to hear the editor's view of the scope and purpose of the GMD journal as we feel that the reviewer has a different interpretation of these to our own understanding, which is that the journal is intended to be a home for documentation of important underpinning science in environmental modelling. Here we are documenting updates to JULES, which as stated in our Introduction is the point of our paper, and we believe this is in line with what is expected for the scope of a Model Development journal.

We will revise the initial presentation to make some of this motivation clearer, and below we provide some more detailed responses to the points in the review.

*Major comments:*

*1. The manuscript does not present a clear modelling concept of how the both processes are presented, in terms of qualitative description and/or supporting it by a flow-chart which would also guide the reader through the manuscript.*

We would be happy to include a brief overview of the processes that will be presented through the course of the paper inserted at the end of the 'Introduction' section to guide the reader more clearly through the rest of manuscript.

*2. The fact that land-use can be regarded as a disturbance is flawed because land-use change is a permanent, very often irreversible change in land-cover. It is reversible when people abandon their fields and that depends on socio-economic conditions that motivate human decision-making. Such a reflection is missing in the introduction.*

We disagree that referring to land-use as a disturbance is flawed, as many other published studies refer to land-use as a disturbance, including the one that the reviewer is presumably citing, although the full reference is not provided (Foster et al., 1998). We do refer to agricultural abandonment in the manuscript 'Introduction', however we would be happy to include extra text to make the point that land-use change is often a long-term change, and both agricultural development and abandonment depends on socio-economic conditions and decision making.

*3. The literature overview leaves the reader with an unclear message, other than it is very complex. However, the literature, also the cited does allow to conclude which processes are essential to incorporate land-use and fire disturbance in land surface models such as JULES. The problem statement that DGVMs have to properly consider disturbances has been identified already in papers in, e.g. Foster et al. 1998, and has been implemented in many ways in many DGVMs since then. This applies also to land-use.*

Please see response at the start.

*4. The methods section starts with explaining how the disturbance term is implemented in the major equation on quantifying changes in vegetation. And here starts the problem of the modelling approach: what is presented is a simply cookie-cutter approach to correct PFT coverage by the proportion of fire disturbance and land-use. Such an approach represents the level of science of the 1990ies. Since then many more advanced approaches also simple ones have been published from which this modelling concept can profit.*

Please see response at the start.

*5. The remainder introduction of equations in the methods section is referring to already published modelling studies and the text does not explain how this was adapted to the current model version or what was updated given the latest progress in science in that field. Therefore, I cannot identify any added scientific value in terms of modelling approaches from which other modelling groups would profit.*

As stated throughout the text, the equations describe the addition of new processes of land-use change ($\alpha a_i$) and fire ($\beta_i$,) which are being published here for the first time. We will signpost these additions again ahead of the equations in the text.

**6. Variables in equation 1 are insufficiently defined or explained.**

We will provide an additional summary table of all terms and units used in equations 1-8 within the 'Model description and developments' section.

**7. The feedback of fuel availability on vegetation distribution is not explained (equation 3).**

We have already stated that "Fire disturbance, $\beta_i$, is included as a PFT-dependent burnt area which can vary in space and time" and following equation 3 that "The calculation of burnt area depends on fuel availability, including soil carbon density…". We will add a sentence that explicitly states that "The calculation of burnt area depends on fuel availability as documented in Mangeon et al., (2016) and which now includes the additional feedback of reduction in fuel from fire (equation 3). Also included in fuel availability is soil carbon density, $C_s$, providing additional mechanisms by which fire and land-use can feedback onto vegetation distribution" which we trust now makes this clear.

**8. From this starting point or poor modelling concept and inadequate description, it makes in my view no sense to review the remaining part of the manuscript because it makes it impossible to judge if the results produced are based on solid ground or if they can be reproduced.**

The JULES model code is all freely available, as is the suite used for this analysis, as described at the end of the paper to ensure reproducibility. The changes described in equations 1-8 are all included in the main trunk of JULES from version 4.8, as detailed in the 'Code Availability' section.

References

Foster, D., Motzkin, G. & Slater, B. Ecosystems (1998) 'Land-Use History as Long-Term Broad-Scale Disturbance: Regional Forest Dynamics in Central New England' 1: 96. https://doi.org/10.1007/s100219900008

Friedlingstein, P., P. Cox, R. Betts, L. Bopp, W. von Bloh, V. Brovkin, P. Cadule, S. Doney, M. Eby, I. Fung, G. Bala, J. John, C. Jones, F. Joos, T. Kato, M. Kawamiya, W. Knorr, K. Lindsay, H.D. Matthews, T. Raddatz, P. Rayner, C. Reick, E. Roeckner, K. Schnitzler, R. Schnur, K. Strassmann, A.J. Weaver, C. Yoshikawa, and N. Zeng, 2006: Climate–Carbon Cycle Feedback Analysis: Results from the C4MIP Model Intercomparison. J. Climate, 19, 3337–3353, https://doi.org/10.1175/JCLI3800.1

Ciais, P., C. Sabine, G. Bala, L. Bopp, V. Brovkin, J. Canadell, A. Chhabra, R. DeFries, J. Galloway, M. Heimann, C. Jones, C. Le Quéré, R.B. Myneni, S. Piao and P. Thornton, 2013: Carbon and Other Biogeochemical Cycles. In: Climate Change 2013: The Physical Science Basis. Contribution of Working Group I to the Fifth Assessment Report of the Intergovernmental Panel on Climate Change [Stocker, T.F., D. Qin, G.-K. Plattner, M. Tignor, S.K. Allen, J. Boschung, A. Nauels, Y. Xia, V. Bex and P.M. Midgley (eds.)]. Cambridge University Press, Cambridge, United Kingdom and New York, NY, USA.

Le Quéré, C., Andrew, R. M., Canadell, J. G., Sitch, S., Korsbakken, J. I., Peters, G. P., Manning, A. C., Boden, T. A., Tans, P. P., Houghton, R. A., Keeling, R. F., Alin, S.,

Andrews, O. D., Anthoni, P., Barbero, L., Bopp, L., Chevallier, F., Chini, L. P., Ciais, P., Currie, K., Delire, C., Doney, S. C., Friedlingstein, P., Gkritzalis, T., Harris, I., Hauck, J., Haverd, V., Hoppema, M., Klein Goldewijk, K., Jain, A. K., Kato, E., Körtzinger, A., Landschützer, P., Lefèvre, N., Lenton, A., Lienert, S., Lombardozzi, D., Melton, J. R., Metzl, N., Millero, F., Monteiro, P. M. S., Munro, D. R., Nabel, J. E. M. S., Nakaoka, S.-I., O'Brien, K., Olsen, A., Omar, A. M., Ono, T., Pierrot, D., Poulter, B., Rödenbeck, C., Salisbury, J., Schuster, U., Schwinger, J., Séférian, R., Skjelvan, I., Stocker, B. D., Sutton, A. J., Takahashi, T., Tian, H., Tilbrook, B., van der Laan-Luijkx, I. T., van der Werf, G. R., Viovy, N., Walker, A. P., Wiltshire, A. J., and Zaehle, S.: Global Carbon Budget 2016, Earth Syst. Sci. Data, 8, 605-649, https://doi.org/10.5194/essd-8-605-2016, 2016.

---

## Referee Report (RR1)

**Review Comments to "Representation of disturbance in the Joint UK Land Environment Simulator Vn4.8 (JULES)"**

**General Comment:**

This study touches on the issue of representation fire disturbance in the Joint UK Land Environment Simulator (JULES). New model features including a better description on updating the natural mortality due to the fire activity/burnt area simulated by INFERNO (Mangeon et al, 2016) for five selected PFTs (plant function types: broadleaf tree needle leaved tree, shrub, C3&C4 grass) and formulations of the carbon emission due to fires from two selected soil carbon pools (decomposable and resistant plant material soil carbon pools) are presented in the manuscript. In the revised manuscript, the authors have been already addressed most of the comments from previous two reviewers.

The behavior of enhanced model was tested with and without imposed the global LULCC (S2&S3), and incorporating the new feature (S2F & S3F). The authors concluded that the new implementation can improve the model representation of the global vegetation cover after considering the global LULCC and fire disturbance simulated by INFERNO, however the new implementation shows the deficit on the tree coverage over the boreal regions. I think this new development is quite important for the applications of the UK ESM, especially for understanding the impact of biomass burning in the future.

Due to the scope of manuscript type is model development for the publication in GMD, I have a few suggestions for the authors to revise their manuscript. In the third section, I suggest to rename the title "Method" to "Experimental set-up and model evaluation", and also to provide a more detail description on the forcing to the developed model, for example: the potential forest/agriculture land cover fraction in HYDE 3.2 from 19XX to 2015 was forced and updated annually to the land for the S3/SF3 simulations, and what is expected to be observed from comparing the result between two simulations. Within this section, I also spotted a few minor issues which were unclear to me, please see the specific comment for the detail. In the result section, the authors only chosen the result for the present day (2014) to evaluate the model performance; however it would be nice to show the mean state from long-term period to avid the model bias in warm year or cold year. Another approach is by adding additional information to the **Fig. 4**, which shows the transient evolution of the model prediction, upper limited and lower limited from the observation.

Finally, I recommend the authors to restructure the discussion session into several sub-sessions, i.e. current model limitation, modelling the disturbance (including the anthropogenic disturbance and natural disturbance), modelling improvement in the future, and others. For example, the model currently only can simulate a realistic fire

activity with a reliable LULCC reconstruction/observation. For the representation of fires, does the model capture the fire activity in the peatland and its emission over the peatland? Fires over those region can produce heavy air pollution and transport pollutants from tropics to temperate climate zone. Does the model reasonable simulate the demography after the fire disturbance? Does the model explicitly couple the other natural disturbance agents, such as windthrow or pest outbreak, through a large scale LULCC forcing after this new development? I listed several references which are relevant for these discussion.

References:

Yue, C., et al.: Representing anthropogenic gross land use change, wood harvest, and forest age dynamics in a global vegetation model ORCHIDEE-MICT v8.4.2, *Geosci. Model Dev.*, 11, 409-428, https://doi.org/10.5194/gmd-11-409-2018, 2018.

Haverd, V., et al.: A new version of the CABLE land surface model (Subversion revision r4601) incorporating land use and land cover change, woody vegetation demography, and a novel optimisation-based approach to plant coordination of photosynthesis, *Geosci. Model Dev.*, 11, 2995-3026, https://doi.org/10.5194/gmd-11-2995-2018, 2018.

Marra, D. M., et al: Windthrow control biomass patterns and functional composition of Amazon forests, *Global Change Biology*, doi:10.1111/gcb.14457.

Chen, Y.-Y., et al.: Simulating damage for wind storms in the land surface model ORCHIDEE-CAN (revision 4262), *Geosci. Model Dev.*, 11, 771-791, https://doi.org/10.5194/gmd-11-771-2018, 2018.

Landry, J.-S, et al.: Modelling long-term impacts of mountain pine beetle outbreaks on merchantable biomass, ecosystem carbon, albedo, and radiative forcing. *Biogeosciences* 13, 5277-5295, 2016.

**Specific Comment:**
1. Please provide a table which summaries parameters that were tuned in this study. For example, in the P30L15, I can't find the information of the spreading parameter (lambda) in the Table 1.
2. Please explain the value of theta (soil moisture) that applies for the equation (4). Does this value represent a vertical average of soil layer or using the soil depth associate with the soil carbon pools for the decomposable and resistant plant material?
3. Can you explain why most of your simulation result from the SF3 shows a relative low tree coverage which comparing with the ESA observation, when you apply a

smaller background mortality (half of the original values) for tree species in the new development?

4. Please add an extra column in **Table 3**, which indicates the variable was updated from LULCC map or INFERNO fire module.

**Technical Comment:**

1. In the **Fig. 5**, please replace "S2, fire" to "SF2, fire" and "S3, fire" to "SF3, fire" in the figure legend for the consistence

2. Please use "burnt area" to replace "burned area" throughout the manuscript both for plots and texts throughout the manuscript.

3. Please use "windthrow" to replace "windfall"

4. When doing a final check of the references cite in this manuscript, I can't find the citation of Avitabile et al. 2016: An integrated pan-tropical… in the text. In the **P17L4**, "Klein Goldewijk et al., 2011" should be "Klein Goldewijk et al., 2013".

---

## Author Response (AR2)

Draft manuscript: 'Representation of fire, land-use change and vegetation dynamics in the Joint UK Land Environment Simulator Vn4.9 (JULES)" submitted to GMDD

We thank the reviewers for taking the time to review our manuscript, and for their helpful comments which we respond to below.

**Anonymous Referee #3**

**p.9, l. 31: I am not sure what you mean with over-disturbance.**

We refer to the changes from fire and land-use together that result in too much disturbance in some regions. We will change the wording to 'too much disturbance' to make this clearer.

**p. 10, l. 1: how do land-use and fire interact?**

We have tried to provide a detailed description of the interaction of land-use and fire in the Introduction; in the model both act as 'disturbances' to vegetation cover, which can be seen in the Results by reducing tree cover and increasing grasses. In reality there are more complex interactions between both processes, which again we have tried to describe in the Introduction, where fire is often used as a method of land-clearance, and land-use can act as both fire ignition and suppression which can vary regionally. The aim of the paper is to describe the development of both fire and land-use as distinct and important 'disturbance' processes within JULES, which can enable further developments of their more complex interactions in the future.

**p.10, l. 12: the new disturbance term is based on the INFERNO model developed by mangeon et al? Please refer to the previous development here.**

Yes, the new disturbance term is based on INFERNO (Mangeon et al., 2016) we which do reference later on in the more detailed description of the developments, and thank you for pointing out that this should also be referenced at the start of the text. We have included the reference at the start of the Introduction now.

**p.11 l. 30: Andela et al. show that the fire occurrence has been reduced, however there is no strong support of fragemented fuel loads in the study. Bistinas et al (2014) also show a reduction of burned area with increases in croplands.**

Thank you for pointing this out. We will revert to the previous text for this, which said 'making fuel less readily available' and also add in the Bistinas et al (2014) reference.

*p.12 l. 1: see Veraverbeke et al. (2017) for the influence of lightning in boreal regions*

Thank you for drawing our attention to this paper, we will include this reference as well.

*p.12 l. 4: delete "these"*

Corrected.

*p.12 l. 5: some references to previous developments in other models could be informative. See for instance Reick et al. (2013)*

We have added in extra references here as suggested (Sitch et al., 2015; Betts et al., 2015; Seo and Kim, 2018).

*p. 13 l. 27: Whether a PFT is dominant or not is described in the previous paragraph? Then this is not the new development? only the definition of cij for the land use is new? I would explain the cij in the previous paragraph and here only describe the adjustment for land use. It is a bit unclear to me what is the new development. Clark et al. (2011) describe a similar procedure not allowing woody PFTs to establish on agricultural lands.*

We have modified the text as suggested. Clark et al. (2011) refer to the original implementation of agricultural land in MOSES (Cox 2001) as used in HadCM3LC (Cox et al., 2000) and used in C4MIP (Friedlingstein et al., 2006). In this implementation agricultural land was fixed in time. The scheme was extended in MOSES to allow for transient land-use change in HadGEM2-ES as part of CMIP5 (Jones et al, 2011). The scheme was then imported into the JULES version of TRIFFID as part of the Global Carbon Budget 2014 (Le Quere et al., 2015). It is this scheme that is described here.

*p. 15, table 2: how did you choose the parameter values?*

These are the same values as those used in the original diagnostic version of INFERNO. We will state this in the text ahead of the table.

*p.18, l. 12: if the LULCC only simulation includes fires in the global disturbance term I would name it "LULCC and constant fire" instead.*

The LULCC only simulation includes the global disturbance term which does not separate out fire, but represents all other disturbances together including fire, pests, windthrow and disease, which we have described in the Methods as "using the standard large-scale constant disturbance rate for the purposes of comparison"

*p. 18 l.26: you use the manhatten metric for the MODIS VCF? the VCF gives the percentage, so you can simply use the NME.*

The Manhattan Metric (MM) is designed for item comparisons, where the variable is a bounded fraction/percentage cover such as vegetation cover, and has been used in previous studies using VCF cover (Kelley et al. 2013; Kelley et al. 2014; Song et al. 2016; Rabin et al. 2017 as just some examples). NME would be inappropriate for two item comparison, and not possible for comparisons with three or more items, and therefore for consistency we use MM for all. In addition, both NME and MM are absolute mean distance measures, so using NME would not give any extra information.

*p.26, l. 15: how do you turn burned area into tree mortality, probably tree mortality is too high? But I actually do not see a strong overestimation of grasses in Congo or Cerrado. Please refer to a figure. also in the previous paragraph, please refer to a figure or table.*

Burned area is directly converted into vegetation mortality, so yes tree mortality may be too high in some cases – this is outlined briefly in the Discussion around variation in species resilience to fire, and could be a key 'next step' development. We have now added in references to figures in these two paragraphs.

*p.30, l. 5: you could also compare to Bond et al. 2005*

Good point, we will include this reference as well.

*p.30, l. 20: the point that reductions in burned area are found with increases in cropland may explain this, see references and paragraph in the introduction.*

We agree. We will add in the reference to Bistinas et al (2014) again within the Discussion.

*p.31 l.3: adaptation to fire and resprouting might be plausible explanations too.*

*I see the last two comments are already adressed in the manuscript, you could probably rephrase the paragraph to indicate earlier that you are offering possibilities to solve the "over-disturbance".*

We have added a sentence near the start of this paragraph to signpost this part of the discussion more clearly: "There are a number of ways this could be addressed in future developments."

*p.31. l.4-5: there is also considerable underestimation of the burned area maxima in the model.*

Agreed. We will note this in the Discussion as well.

*p.31, l.8-9: see Pfeiffer et al. 2013 for inclusion of cropland fragmentation in a fire model.*

Noted. We will include a reference to this method in the Discussion.

*Betts, R. A., N. Golding, P. Gonzalez, J. Gornall, R. Kahana, G. Kay, L. Mitchell & A. Wiltshire Climate and land-use change impacts on global terrestrial ecosystems and river flows in the HadGEM2-ES Earth system model using the representative concentration pathways. Biogeosciences, 12, 1317-1338. 2015.*

*Clark, D., Mercado, L., Sitch, S., Jones, C., Gedney, N., Best, M., Pryor, M., Rooney, G., Essery, R., Blyth, E., Boucher, O., Harding, R., Huntingford C., & Cox, P.: The Joint UK Land Environment Simulator (JULES), model description - Part 2: Carbon fluxes and vegetation dynamics, Geoscientific Model Development, 4, 701-722, 2011.*

*Cox, P., Betts, R., Jones, C., Spall, S., & Totterdell, I.: Acceleration of global warming due to carbon-cycle feedbacks in a coupled climate model, Nature, 408, 184-187, 2000.*

*Cox, P.M.: Description of the "TRIFFID" Dynamic Global Vegetation Model, Tech. Note 24, Hadley Centre, Met Office, 16 pp. 2001.*

*Friedlingstein, P., P. Cox, R. Betts, L. Bopp, W. von Bloh, V. Brovkin, P. Cadule, S. Doney, M. Eby, I. Fung, G. Bala, J. John, C. Jones, F. Joos, T. Kato, M. Kawamiya, W. Knorr, K. Lindsay, H.D. Matthews, T. Raddatz, P. Rayner, C. Reick, E. Roeckner, K. Schnitzler, R. Schnur, K. Strassmann, A.J. Weaver, C. Yoshikawa, and N. Zeng: Climate–Carbon Cycle Feedback Analysis: Results from the C4MIP Model Intercomparison. J. Climate, 19, 3337–3353, https://doi.org/10.1175/JCLI3800.1, 2006.*

*Jones, C., Hughes, J., Bellouin, N., Hardiman, S., Jones, G., Knight, J., Liddicoat, S., O'Connor, F., Andres, R., Bell, C., Boo, K., Bozzo, A., Butchart, N., Cadule, P., Corbin, K., Doutriaux-Boucher, M., Friedlingstein, P., Gornall, J., Gray, L., Halloran, P., Hurtt, G., Ingram, W., Lamarque, J., Law, R., Meinshausen, M., Osprey, S., Palin, E., Chini, L., Raddatz, T., Sanderson, M., Sellar, A., Schurer, A., Valdes, P., Wood, N., Woodward, S., Yoshioka, M., & Zerroukat, M.: The HadGEM2-ES implementation of CMIP5 centennial simulations, Geoscientific Model Development, 4, 543-570, 2011.*

*Kelley DI, Prentice IC, Harrison SP, Wang H, Simard M, Fisher JB, Willis KO. A comprehensive benchmarking system for evaluating global vegetation models. Biogeosciences. May 17;10(5):3313-40. 2013.*

*Kelley DI, Harrison SP, Prentice IC. Improved simulation of fire-vegetation interactions in the Land surface Processes and eXchanges dynamic global vegetation model (LPX-Mv1). Geoscientific Model Development. Oct 16;7(5):2411-.2014.*

*Le Quéré, C., Moriarty, R., Andrew, R. M., Peters, G. P., Ciais, P., Friedlingstein, P., ... Zeng, N. (2015). Global carbon budget 2014. Earth System Science Data, 7(1), 47–85. https://doi.org/10.5194/essd-7-47-2015.*

Sitch, S., Friedlingstein, P., Gruber, N., Jones, S., Murray-Tortarolo, G., Ahlstrom, A., Doney, S., Graven, H., Heinze, C., Huntingford, C., Levis, S., Levy, P., Lomas, M., Poulter, B., Viovy, N., Zaehle, S., Zeng, N., Arneth, A., Bonan, G., Bopp, L., Canadell, J., Chevallier, F., Ciais, P., Ellis, R., Gloor, M., Peylin, P., Piao, S., Le Quéré, C., Smith, B., Zhu Z., & Myneni, R.: Recent trends and drivers of regional sources and sinks of carbon dioxide, Biogeosciences, 12, 653-679, 2015.

Seo, H. and Kim, Y.: Interactive Impacts of Fire and Vegetation Dynamics on Global Carbon and Water Budgets using Community Land Model version 4.5, Geosci. Model Dev. Discuss., https://doi.org/10.5194/gmd-2018-231, in review, 2018.

Song, Xiang, et al. "Development of an establishment scheme for a DGVM." Advances in Atmospheric Sciences 33.7: 829-840. 2016.

Rabin SS, Melton JR, Lasslop G, Bachelet D, Forrest M, Hantson S, Kaplan JO, Li F, Mangeon S, Ward DS, Yue C. The Fire Modeling Intercomparison Project (FireMIP), phase 1: experimental and analytical protocols with detailed model descriptions. Geoscientific Model Development. Mar 17;10(3):1175-97. 2017

**Anonymous Referee #4**

**General Comment:**

*Due to the scope of manuscript type is model development for the publication in GMD, I have a few suggestions for the authors to revise their manuscript. In the third section, I suggest to rename the title "Method" to "Experimental set-up and model evaluation", and also to provide a more detail description on the forcing to the developed model, for example: the potential forest/agriculture land cover fraction in HYDE 3.2 from 19XX to 2015 was forced and updated annually to the land for the S3/SF3 simulations, and what is expected to be observed from comparing the result between two simulations.*

We have updated the Method section title, and provided more detail of the land-use forcing as suggested.

*Within this section, I also spotted a few minor issues which were unclear to me, please see the specific comment for the detail. In the result section, the authors only chosen the result for the present day (2014) to evaluate the model performance; however it would be nice to show the mean state from long-term period to avid the model bias in warm year or cold year. Another approach is by adding additional information to the Fig. 4, which shows the transient evolution of the model prediction, upper limited and lower limited from the observation.*

We have now updated the results to represent a five year mean 2010-2015 across all plots. Transient evolution is not available in CCI and unassessed for VCF observations so the alternative suggested approach would not be possible.

*Finally, I recommend the authors to restructure the discussion session into several sub-sessions, i.e. current model limitation, modelling the disturbance (including the anthropogenic disturbance and natural disturbance), modelling improvement in the future, and others. For example, the model currently only can simulated a realistic fire activity with a reliable LULCC reconstruction/observation. For the representation of fires, does the model capture the fire activity in the peatland and its emission over the peatland? Fires over those region can produce heavy air pollution and transport pollutants from tropics to temperate climate zone. Does the model reasonable simulate the demography after the fire disturbance? Does the model explicitly couple the other natural disturbance agents, such as windthrow or pest outbreak, through a large scale LULCC forcing after this new development? I listed several references which are relevant for the discussion.*

We have now included sub-headings to signal different sections within the Discussion section.

The model does not yet capture peatland fires, which was noted in Mangeon et al. (2016). We agree that this is an important point for future development considering the large-scale fire emissions from peatland areas, but we believe this would be more appropriately discussed within a paper focused on fire emissions rather than vegetation cover, where more examples and evidence could be provided for this development requirement. The slow recovery time within TRIFFID is also a known issue which needs further work, as mentioned in the Discussion.

As stated on page 25 line 16 (using line numbers shown below), "We have not considered windthrow, pests, and diseases etc., which for now are still aggregated into the generic large-scale disturbance term in JULES."

*Specific Comment:*

*1. Please provide a table which summaries parameters that were tuned in this study. For example, in the P30L15, I can't find the information of the spreading parameter (lambda) in the Table 1.*

We have updated Table 1 to include the values for lambda before and after tuning.

*2. Please explain the value of theta (soil moisture) that applies for the equation (4). Does this value represent a vertical average of soil layer or using the soil depth associate with the soil carbon pools for the decomposable and resistant plant material?*

Theta is the unfrozen soil moisture as a fraction of saturation in the top soil layer (0-10cm), where much of the soil carbon is expected to be located. Note that the assumption that soil carbon is located close to the surface is also made by the model's soil respiration calculation, which also uses top soil layer moisture. Theta has been more clearly defined in the text.

*3. Can you explain why most of your simulation result from the SF3 shows a relative low tree coverage which comparing with the ESA observation, when you apply a smaller background mortality (half of the original values) for tree species in the new development?*

As outlined in the Discussion, this is likely due to slow recovery time within TRIFFID, too much disturbance from including both fire and land-use, too high burnt area / mortality rate, not excluding cropland areas from burning, and / or requirement for scaling by PFT for fire mortality to represent fire resilience. There is a slight increase in tree coverage in the central Amazon and high latitude North Russia where burnt area is low, that may be a result of the reduced background mortality (Fig 3).

*4. Please add an extra column in Table 3, which indicates the variable was updated from LULCC map or INFERNO fire module.*

We have included an extra column to signify the source of each variable.

*Technical Comment:*

*1. In the Fig. 5, please replace "S2, fire" to "SF2, fire" and "S3, fire" to "SF3, fire" in the figure legend for the consistence*

Corrected.

*2. Please use "burnt area" to replace "burned area" throughout the manuscript both for plots and texts throughout the manuscript.*

Corrected.

*3. Please use "windthrow" to replace "windfall"*

Corrected.

*4. When doing a final check of the references cite in this manuscript, I can't find the citation of Avitabile et al. 2016: An integrated pan-tropical… in the text. In the P17L4, "Klein Goldewijk et al., 2011" should be "Klein Goldewijk et al., 2013".*

Corrected.

[revised manuscript text omitted]